# Experimental Hybridization in *Leishmania*: Tools for the Study of Genetic Exchange

**DOI:** 10.3390/pathogens11050580

**Published:** 2022-05-14

**Authors:** Tiago R. Ferreira, David L. Sacks

**Affiliations:** Intracellular Parasite Biology Section, Laboratory of Parasitic Diseases, National Institute of Allergy and Infectious Diseases, NIH, Bethesda, MD 20892, USA; tiago.rodriguesferreira@nih.gov

**Keywords:** *Leishmania*, hybridization, meiosis, sand flies, genetic exchange

## Abstract

Despite major advances over the last decade in our understanding of *Leishmania* reproductive strategies, the sexual cycle in *Leishmania* has defied direct observation and remains poorly investigated due to experimental constraints. Here, we summarize the findings and conclusions drawn from genetic analysis of experimental hybrids generated in sand flies and highlight the recent advances in generating hybrids in vitro. The ability to hybridize between culture forms of different species and strains of *Leishmania* should invite more intensive investigation of the mechanisms underlying genetic exchange and provide a rich source of recombinant parasites for future genetic analyses.

## 1. Introduction

Kinetoplastid parasites of the *Leishmania* genus comprise a wide diversity of species, with genetically distinct populations causing a variety of human and zoonotic diseases collectively referred as the leishmaniases; an estimated one billion people are at risk of being infected. Infections can lead to a wide spectrum of clinical presentations ranging from self-healing cutaneous lesions to visceralization and death, largely dependent on the species of parasite and the host’s immunogenetic background. The well-described heteroxenous life cycle of *Leishmania* involves transmission of promastigote forms through the bite of a phlebotomine sand fly insect to the mammalian host skin, where they differentiate into amastigotes inside phagocytic immune cells [1].

The question of how *Leishmania* reproduce remains a matter of debate, in large part because while asexual reproduction is readily apparent among life cycle stages found both in the sand fly vector and within phagocytes in the mammalian host, direct observation of sexual reproduction has thus far not been possible. This fact, along with a number of observations based on population genetics, support a predominant if not strictly clonal reproductive strategy [2]. In particular, the high linkage disequilibrium observed among different polymorphic alleles is thought to be due to the absence of mechanisms, namely gene segregation and recombination, which tend to disrupt the common inheritance of alleles on the same or different chromosomes [3]. Population genetic studies, however, have been used to provide evidence for sex, whereby multi locus genotyping and whole genome sequencing analysis of natural isolates have identified hybrids between strains of closely related species and even between divergent species [4,5,6,7,8]. The first experimental demonstration that *Leishmania* strains can hybridize in a manner consistent with meiotic sex was provided by Akopyants et al. in 2009 [9], in which two recombinant lines of *Leishmania major* each bearing a different antibiotic resistance gene were used to co-infect sand flies, followed by recovery and cloning of midgut promastigotes that were resistant to both drugs. The hybrid progeny had inherited multiple alleles from both parents. These studies formally demonstrated that *Leishmania* possess the machinery for genetic exchange and identified promastigotes in the sand fly midgut as a life cycle stage in which hybridization can occur. Following this initial report, experimental intraspecific, interspecific, and intraclonal hybrids have been recovered from or observed in sand flies co-infected with recombinant lines of *Leishmania* containing different antibiotic resistance and/or fluorescence genes [10,11,12,13,14]. Most recently, both ourselves and others have reported that axenic culture forms of promastigotes can form stable hybrids entirely in vitro [15,16,17]. Possible hybrids formed between intracellular amastigotes residing in a common macrophage vacuole have been described as well [18]. Thus, the continuing debate regarding clonality versus sexuality in *Leishmania* pertains mainly to the frequency of hybridization, the mode of genetic exchange (whether there a true sexual cycle), and the degree to which recombination has been able to break the prevalent pattern of a clonal population structure [19]. 

It is important to point out that in the absence of direct observations of cell fusion, nuclear fusion, and meiosis producing haploid gametic cells, conclusions regarding a true sexual cycle in *Leishmania* based on observations of hybrid genotypes can only be inferred. This review will focus on the biology of the hybrids which have been generated experimentally and which have been analyzed by high resolution whole-genome sequencing in order to inform the mode of genetic exchange. We summarize the approaches current being used to dissect the involvement of meiotic gene orthologs in genetic exchange and to identify and select for mating competent cells. Finally, the potential of experimental hybridization for linkage studies and positional cloning of important genes is critically discussed.

## 2. Experimental *Leishmania* Hybrids in Sand Flies

Formal demonstration of genetic exchange in *Leishmania* spp. during their growth and development in the sand fly vector has been provided by different research groups, and it is widely accepted that inter- and intraspecific hybrids can be generated experimentally in sand flies. The lack of availability of colonized sand flies to most *Leishmania* research programs and the low frequency at which these hybridizations take place in colonized flies represents a major challenge in the field. In experimental designs where parental lines have a single allele replaced by a selectable marker, only 25% of possible mating events are tracked. In addition, frequencies of hybrid recovery from laboratory sand flies are highly variable and species-dependent (both *Leishmania* and sand fly). Double drug resistant hybrids have been recovered in 7–26% of infected sand flies, either *Phlebotomus duboscqi* (natural *L. major* vector) or *Lutzomyia longipalpis* (permissive vector) for intraspecies *L. major* crosses, 34–65% in *L. tropica*, and 3.2% for inter-species *L. major* × *L. infantum* [11,12,13]. The relatively low frequency of hybrid recovery may be related to the use of culture-adapted recombinant parasites for hybrid selection, and it is unknown whether fresh clinical isolates might hybridize with greater efficiency in the vector.

Genetic exchange analyses thus far have relied on the introduction of two different drug resistance markers in each of the parental strains to allow for selection of double drug resistance hybrids. This is an important constraint of the current experimental model, as hybrids must stably grow as promastigotes in the presence of both selection antibiotics for several generations. The use of parental lines that overexpress fluorescent proteins with discernable spectra can provide quick PCR-free initial confirmation of hybridization using flow cytometry analysis or confocal microscopy. However, co-expression of parental fluorescent proteins requires additional confirmation. It is critical to be able to distinguish between a product of genetic exchange and an event of cytoplasmic content sharing. Exchange of flagellar and cytosolic proteins has been shown in *T. brucei* via transient flagellar membrane fusion or extracellular vesicle endocytosis to give rise to trypanosomes carrying exogenous proteins stable for up to three days [20,21]. Although protein exchange can occur between *T. brucei* haploid cells, there is no transfer of genetic material during this process [20]. Cell-to-cell communication is an underexplored subject in *Leishmania* research, and it is possible that it takes place inside restricted compartments, such as in the sand fly thoracic midgut, or intraphagosomally in the case of *L. mexicana* or *L. amazonensis*, which share the same host cell vacuole. Single-cell spatial genomic analysis of mixed infections is a potential way forward to acquiring in situ information and avoiding the requirement of parasite growth in culture.

Analysis of parental SNP inheritance has been reliably used for tracking biallelic segregation in *Leishmania* genetic exchange in several experimental crosses (Figure 1). It has been demonstrated by multilocus genotyping methods ranging from PCR amplification followed by Sanger sequencing, SNP–cleaved amplification polymorphic site (SNP-CAPS) analysis, Southern Blotting, and bulk Whole Genome Sequencing (bWGS) [9,11,22]. In 2009, PCR-based approaches were used to confirm the first demonstration of hybrid generation in laboratory sand flies between two *L. major* strains, namely, Fn and LV39 [9]. All of the eighteen *L. major* hybrids showed biparental inheritance of the seven chromosomes tested. These findings were extended to 96 additional hybrids generated in sand flies, involving different pairwise combinations of *L. major* strains obtained from across the geographic range of this species [11]. Subsequently, the first experimental inter-species hybrids were described, with eleven hybrids found to inherit both *L. major* and *L. infantum* parental alleles in all but one of the chromosomes tested [13]. Thus, there is no evidence to date for mating types that might limit genetic exchange; experimentally, at least, there is no species barrier to mating between strains. Importantly, attempts to recover double drug resistant hybrids following co-infection of mice using the same recombinant *L. major* parental lines that hybridized in the vector were unsuccessful [9], suggesting that the sand fly is where sex happens. The descriptions of possible in vitro hybridization involving amastigotes of the *L. mexicana* complex residing in a communal vacuole inside macrophages, while extremely interesting, could not be genetically validated because of instability in the growth of the double drug resistant lines [18]. In these studies, co-infection of mice with the parental lines failed to yield recoverable hybrids. 

The initial genotyping of the experimental hybrids provided evidence that they were full-genome hybrids based on their bi-parental inheritance of the limited number of markers tested. Subsequently, all of the 36 chromosomes in the genomes of the intra- and interspecific hybrids were investigated in detail using Illumina short-read bWGS [12]. Genome-wide analysis of the SNPs that are homozygous and different between the parents revealed that 99% of the disomic chromosomes in the progeny had a parental-derived SNP frequency close to 0.5, inferring a near full genome hybridization with the allele contributions from each parent matching the inheritance patterns expected under meiosis. The bWGS analysis of the parents and hybrid clones revealed the copy number of the individual chromosomes and the patterns of somy inheritance. While *Leishmania* are largely diploid, all *Leishmania* strains show varying degrees of aneuploidy, including mosaic aneuploidy, which refers to somy variations that are present within clonal populations [23]. Thus, the analysis of somy at the population level, as in the published studies, computed average values. In almost every case, chromosomes that were present in close to three copies in either parent were transmitted to the hybrid progeny in either one or two copies and in frequencies expected under meiosis. Finally, bWGS analysis of *L. major* backcross progeny clones provided the first clear evidence for genome-wide recombination events in *Leishmania*, demonstrating that classical crossing over occurs at meiosis. In the absence of direct observation of haploid gametic cells, the inheritance patterns revealed by WGS analysis in experimental hybrids remain the strongest argument in favor of meiosis-like sexual recombination in the *Leishmania* genus. 

The conclusions regarding a meiosis-like program controlling genetic exchange in *Leishmania* is complicated to a certain extent by the unexpected patterns of chromosome inheritance observed in hybrid progeny. The exceptions to Mendelian ratios included chromosomes that showed uniparental inheritance and changes in chromosome copy numbers observed either at the level of individual somy or whole ploidy. Thus, 2% of the chromosomes in the *L. major* intra-specific hybrids were trisomic despite the chromosomes being disomic in both parents, and 0.7% of the chromosomes showed alleles from only one parent, for which a loss of heterozygosity was inferred [12]. More common to the experimental hybrids was polyploidy, which mirrors findings in *T. brucei*, for which triploid hybrids are a familiar product of experimental matings [24]. Out of 53 *L. major* intra-specific hybrids described between different diploid parental combinations, 39 were diploid, 13 were triploid and one was tetraploid [9,11]. Similar variation was shown for inter-species hybrids. From eleven hybrids between *L. major* and *L. infantum*, five were diploid, five were triploid, and one was tetraploid [13]. The gene dosage difference due to chromosome copy number variation was reflected in the phenotypic segregation observed, with the hybrids that acquired their extra genome from *L. major* producing skin lesions and no dissemination, while the hybrids that acquired their extra genome from *L. infantum* produced strong dissemination to and growth in the viscera no skin pathology. This suggests that tissue tropism is a heritable trait and a potential target for linkage analysis, as discussed in detail below. 

Polyploidy is an especially prevalent phenomenon for in vitro generated hybrids. Out of 24 *L. tropica* in vitro hybrids generated between two diploid parental lines, five were diploid, fifteen were triploid, and four were tetraploid [17]. All three in vitro hybrids from a diploid *L. donovani* × *L. major* cross were triploid, and two in vitro hybrids obtained from a diploid *L. donovani* × triploid *L. tropica* cross were tetraploid [15]. In the case of a recently published method of irradiation-facilitated in vitro hybridization (discussed below), almost all of the progeny were tetraploid [16]. Although classical meiosis entails the production of haploid gametic cells after reductional division and subsequent fusion of two different gametes, alternative models can be proposed to explain the generation of polyploid hybrids. The generation of triploid progeny may involve the production of haploid gametes in one parent and the failure of meiosis in the other parent such that diploid and haploid cells fuse, as has been proposed for *T. brucei* [24]. The formation of tetraploid progeny would suggest the fusion of diploid cells, possibly unreduced gametes, or an aberrant form of hybridization that is unrelated to a meiotic program in either parent. It is possible that the polyploid hybrids reflect an unconventional mode of genetic exchange, such as the parasexual process that occurs in certain fungi [25].

## 3. Kinetoplast DNA Inheritance

Genome-wide analyses revealed biallelic inheritance of the parental nuclear genomes in *Leishmania* F1 progeny for several different crosses performed experimentally (Figure 1). Conversely, inheritance of *Leishmania* extranuclear DNA present in the single mitochondrion (kinetoplast DNA or kDNA) has long been considered uniparental, similar to what is seen in late-branching eukaryotes [26]. Recent evidence suggests a different scenario, with components of the kDNA network showing either biparental or uniparental contribution in experimental and putative natural hybrids [8,15].

The *Leishmania* kDNA is comprised of two main elements: dozens of ~20-kb maxicircles (20–50 molecules in *L. tarentolae*) and thousands of ~0.9-kb minicircles (10,000–20,000 molecules in *L. tarentolae*) concatenated together in a mesh network of circular DNA [27]. Maxicircles are the eukaryotic mitochondrial DNA equivalent, carrying eighteen protein-coding genes involved in the respiratory chain and mitoribosome structure, one 12S and one 9S mitochondrial rRNAs. Each minicircle encodes a guide RNA (gRNA) that targets mitochondrial mRNAs in a highly specialized post-transcriptional regulatory mechanism. Twelve of the eighteen maxicircle-encoded mRNAs require the addition or deletion of uridines (U) for expression in an RNA editing process involving the annealing of gRNAs to the target transcript and the activity of an RNA editing core complex [27,28,29]. 

The inheritance patterns of the full set of genetic components of the kDNA have until recently been under-explored. The use of parental maxicircle polymorphisms in inter-strain crosses as a proxy for overall kDNA inheritance in the early analyses contributed to the inaccurate inference of uniparental kDNA segregation [9]. In fact, recent evidence in *T. brucei* and *Leishmania* suggests that while maxicircle contribution is uniparental, minicircle inheritance is biparental [8,30]. The same has been observed for in vitro *Leishmania* hybrids [15,16]. It is proposed that maxicircle composition in hybrids is derived from both parents upon completion of genetic exchange, while rapid DNA segregation in the following mitotic divisions may lead to the observed loss of contribution by one of the parents. Such loss of complexity is observable in minicircles [8,31], although the effects are less drastic as these are present in thousands of copies and hundreds of minicircle classes (MSCs). MSCs are groups of highly similar minicircle molecules distinguishable within the same *Leishmania* population by 1–5% sequence mismatches (95–99% sequence identity) [32]. Recent findings confirm the long-held notion of minicircle copy number plasticity and the loss of non-essential minicircles in cultured trypanosomatids due to the lack of specific environmental pressure [31]. Genetic exchange would presumably play a significant role in restoring gRNA complexity, preventing the loss of essential genes, and maintaining correct mitochondrial gene expression and metabolism throughout the parasite’s life cycle [8].

Mito-nuclear discordance is therefore a relevant signature of genetic exchange in trypanosomatids, and its detection provides a marker for past hybridization events in interspecific *Leishmania* hybrids in nature [32,33]. Extranuclear DNA inheritance in other protists such as *Plasmodium* is maternal, as male gametes typically lack both plastids and mitochondria [34], similar to products of meiosis in vertebrates. The evidence of parental mitochondrial fusion and the lack of demonstrated heterogamy in *Leishmania* and *T. brucei* support the hypothesis that meiosis produces morphologically indistinguishable sexual stages (isogamy) in trypanosomatids. The in vitro hybridization methods recently described in *Leishmania* (discussed in the next section) may potentially provide a feasible way to study the mechanisms that orchestrate kinetoplast fusion and the kinetics of maxicircle/minicircle segregation. In addition, recent efforts have generated specific bioinformatics packages in R and Python to support the analysis of trypanosomatid minicircle composition from next-generation sequencing data [31,32].

It is unclear whether a more homogeneous minicircle composition could be detected in *Leishmania* inside the sand fly midgut or in infected host samples. Minicircle diversity is most likely not uniform in the cultured *Leishmania* promastigote population, as MSCs are expected to change after successive rounds of mitosis in a stochastic manner. For the study of nuclear genome heterogeneity, single-cell WGS (scWGS) has been used to describe the rise of different nuclear karyotypes within a clonal *L. donovani* culture [35]. Similarly, analysis of single-cell kDNA or single-kinetoplast composition is important for a comprehensive understanding of minicircle heterogeneity, MSC cooperative functioning, and maxicircle–minicircle combinations in *Leishmania* populations.

## 4. Novel Tools for Generating Sand Fly-Free *Leishmania* Hybrids

### 4.1. Leishmania In Vitro Mating

Sexual reproduction in *Leishmania* appears to be a non-obligatory part of its transmission cycle, and it is unknown what triggers the commitment to diverge from the mostly unidirectional asexual promastigote growth and differentiation program to a sexual stage in the insect vector. A major obstacle is the availability of laboratory sand fly colonies, which are restricted to a handful of research groups worldwide. Because sand fly rearing is laborious and resource-consuming, the in vitro hybridization methods recently reported may become a helpful tool for dissecting the cryptic *Leishmania* sexual cycle. In 2020, in vitro generation of bona fide diploid *Leishmania* full genomic hybrids was confirmed for the first time [17]. The most critical factor was the use of *L. tropica* as the parental species. As far as we are aware, these are the only crosses in which *Leishmania* diploid hybrids have been generated by in vitro mating. A second study has recently confirmed the comparatively high mating competency of *L. tropica* in vitro using a different strain [15]. It is unclear why *L. tropica* promastigotes have a remarkably high capacity to undergo experimental genetic exchange both in vitro and in vivo (in 34–65% of infected sand flies) [11]. The experimental mating efficiency seems to reflect the elevated genomic diversity of *L. tropica* isolates and the prevalence of described natural hybrids from Asia carrying an *L. tropica* haplotype [36,37,38,39] Nevertheless, the hybridization rate of *L. tropica* in vitro is significantly lower than in sand flies, with an estimated minimum frequency of hybridization in the order of magnitude of 10^−8^ vs. 10^−4^, respectively. 

In our hands, *L. tropica* was the only mating-competent species in vitro that did not require exposure to DNA stress conditions. However, in a herculean effort, Gutierrez-Corbo et al. reported the recovery of two *L. donovani* × *L. tropica* and three *L. donovani* × *L. major* hybrids after testing 3072 culture wells for each cross in different conditions [15]. The in vitro conditions included using promastigotes in different growth phases and culturing the promastigotes in the presence of an insect feeder cell line derived from the sand fly *Lutzomyia longipalpis*. As the hybridization frequencies were extremely low and not reproducible in independent experiments, it was not possible to draw conclusions from these studies regarding the culture conditions that might promote hybridization. 

### 4.2. DNA Stress as a Trigger for Leishmania Mating In Vitro

In 2022, a modified protocol provided a reproducible enhanced recovery rate of in vitro hybrids by several times over through the treatment of parental *Leishmania* cells with low levels of DNA-damaging agents prior to co-culture [16]. The rationale for these studies was that in organisms that are known to be facultatively sexual, they can be triggered to initiate their sexual cycle by conditions that produce DNA damage [40,41,42]. In our studies, DNA stress was induced by oxidative damage with hydrogen peroxide, by DNA breaks with methyl methanesulfonate (MMS), or by DNA breaks by ionizing gamma radiation. In contrast to the previous approach, the generation of irradiation-facilitated hybrids (referred to here as iHybrids) is a species-agnostic method (*L. major* being the exception thus far), supporting the feasible exploration of meiotic gene function in axenic *Leishmania* culture.

*L. tropica* exposed to 6.5 Gy of gamma-radiation revealed an increase in hybrid recovery of >100-fold in vitro [16]. The iHybrid approach allowed for the in vitro generation of intraspecific hybrids of *L. braziliensis* and *L. donovani* as well as interspecific hybrids between *L. tropica* and *L. infantum* (Figure 1). Trypanosomatids are highly tolerant to gamma radiation, and viable parasites can be recovered after exposures of 500 Gy [43]. Stress conditions inside the vector and mammalian host may have contributed to the selection of pathways to withstand DNA damage in these parasites. In the sand fly midgut, *Leishmania* are exposed to different sources of DNA stress. Blood meal hemoglobin digestion releases heme, which can then be hydrolyzed by heme oxygenase to free Fe^2+^, leading to reactive oxygen species (ROS) and oxidative DNA damage [44]. We speculate that ROS levels in the digested blood meal below a threshold level that is lethal for the parasites or sand fly [45] may be able to trigger repairable DNA damage and genetic exchange. Alternatively, spontaneous DNA breaks are often associated with replicative stress [46]. In the first two days post-blood meal, the sand fly midgut contains proliferating promastigotes marked by DSBs catalyzed during the S phase. This is consistent with the finding that the frequency of in vitro hybridization has thus far been found to peak when the co-cultures are initiated with parental lines in the exponential growth phase.

The major repair mechanism of DNA double-strand breaks (DSB) involves the non-homologous end joining (NHEJ) pathway, which ligates the ends of the break in order to fully restore it. It is widely accepted that trypanosomatids lack a canonical NHEJ and that homologous recombination (HR) is instead the most important DSB repair pathway, although the mechanisms remain unclear (see review in [46]). The drastic increase in in vitro hybridization post-irradiation suggests an interplay between the DNA break repair machinery and meiosis in these parasites, ultimately leading to upregulation of genes involved in the fusion of gametes and nuclei.

### 4.3. In Vitro Hybridization Provides Novel Insights into Leishmania Mating

Endogenous programmed DSBs comprise the first step in a process that initiates meiotic recombination between homologous chromosomes in eukaryotes in general [47]. At the start of the eukaryotic meiotic prophase I, a zipper-like multiprotein structure named the synaptonemal complex (SC) is formed containing a central element and an axial (lateral) element. The latter includes meiosis-specific factors such as Hop1 (HORMA domain-containing protein 1), Red1, and Rec8/Rad21 for homologous chromosome pairing. Spo11 is the effector topoisomerase-like protein in the DSB machinery, which cleaves the genomic DNA at non-random spots and remains covalently bound to both DNA ends until processing by the Mre11-Rad50-NBS1 complex. The DNA-binding protein RPA covers the single-strand ends, and is later replaced by recombinases Dmc1 (disrupted meiotic cDNA 1) and Rad51 (radiation-sensitive 51), which invade the homologous chromosome and form a D-loop structure mediated by the Hop2-Mnd1 (homologous-pairing protein 2—meiotic nuclear division protein 1) complex [48]. The chromosome synapsis is stabilized by Hop2-Mnd1, which inhibits the interaction between non-homologous chromosomes. 

Orthologs of all the meiosis-related proteins referenced here have been found in the *Leishmania* genome [49,50]. Whether or not they retain their conventional meiosis-specific functions in these parasites is unclear. Thus far, Rad51expression has been shown to be upregulated in *L. tropica* promastigotes during irradiation-induced mating in vitro and in *T. cruzi* cell-fused hybrids [16,43]. Apart from the well-conserved components mentioned here, it has been a challenging task to identify homologues of all the meiotic machinery in *Leishmania*. Recent in silico analyses have identified the putative trypanosomatid orthologs of SYCPs (synaptonemal complex proteins), structural components of the vertebrate axial element of the SC. Interestingly, the highest similarities were found between SYCP2-SYCP3 and KKT16-KKT18 proteins, which form the kinetoplastid kinetochore responsible for the mitotic segregation of sister chromatids. It has been proposed that the ancestral kinetoplastid kinetochore was repurposed from meiotic SC structural components [51].

At the end of prophase I, the Spo11-catalyzed DSB is repaired upon recombination and crossover events are resolved. In protists such as *T. brucei*, for which classical gametic meiosis has been observed [52], following a pre-meiotic S phase, meiosis I progresses until segregation of chromosome homologs into two haploid 1n2c daughter cells. If both cells enter classical meiosis II, equational division culminates in the generation of four haploid daughter gametic cells (1n1c). The evidence that polyploid experimental *Leishmania* hybrids are formed along with diploids, particularly in the iHybrid progeny, raises the possibility of two different mechanisms of genetic exchange in *Leishmania,* one of which does not involve meiosis I; however, definitive evidence that polyploid hybrids are a result of incomplete meiosis or of a non-meiotic process such as that described for *Candida albicans* parasexual mating remains lacking [25]. In addition to the difficulty of identifying many homologues of eukaryotic meiotic components in *Leishmania* by their primary amino acid sequence similarity, the lack of a high-throughput experimental model to test null mutants has been a persistent challenge. 

The DNA damage-induced approach is not an exclusive feature of trypanosomatids. DSB causing agents have been described as initiating meiosis in multiple microorganisms and partially replacing Spo11 function [40,53,54]. In social amoebas such as dictyostelids, an unknown mechanism completely independent of Spo11 triggers meiotic recombination [55]. The generation of *Leishmania* iHybrids may bypass the requirement of endogenous Spo11 activity as a trigger of meiosis; thus, evaluation of gene functions directly related to the meiotic DSB machinery may not be appropriate using this method. However, it seems fitting for functional analysis of genes that have a role in later steps such as gamete fusion. 

Regardless of the precise mechanism involved, gametic cell membrane fusion (plasmogamy) and nuclei fusion (karyogamy) in protists, fungi, and other microorganisms are marked by the expression of Hap2/Gcs1 and Gex1/Kar5 genes, respectively. Hap2/Gcs1 (Hapless 2/Generative cell specific 1 protein) is an ancestral gamete fusogen found in non-vertebrates which has been widely used as a marker for gametic cells, including *Plasmodium* and *T. brucei* [56,57]. Its structure resembles that of flavivirus class II membrane fusion proteins [58]. Gex1/Kar5 (gamete expressed 1/karyogamy protein 5) plays a central role in nuclear envelope fusion during budding yeast mating [59]. Single-cell RNA-sequencing (scRNA-seq) identified Hap2 and Gex1 upregulated in a discrete cell population induced by irradiation treatment of *L. tropica* parental strains [16]. Sorting of promastigotes expressing a Hap2-mNeonGreen fusion led to the isolation of both mating-competent (Hap2^+^ cells) and mating-incompetent (Hap2^−^) promastigotes from axenic culture. Experimental evidence of the implication of other specific genes directly in meiotic recombination using this in vitro model will hopefully help to shed light on the sexual reproduction mechanism in *Leishmania*.

## 5. Quantitative Traits and Gene Linkage Analysis 

Apart from the potential of in vitro hybridization to inform the underlying mechanisms of genetic exchange in *Leishmania*, the ability to easily generate large numbers of recombinant parasites holds enormous promise for future genetic analyses seeking to identify the genes controlling important traits. Reverse genetic approaches, in which a gene of interest can be specifically targeted for deletion or overexpression, have thus far had only modest success in identifying the strain- and species-specific genes controlling such critical characters as host range, tissue tropism, and pathogenicity. Future genetic studies offer an unbiased approach to finding causative associations between DNA polymorphisms and phenotypic traits using bi-parental quantitative trait loci (QTL) mapping. QTL mapping can reveal the network of genes that control a complex phenotype. Quantitative traits are frequently more easily examined when gene sequences are isolated from other genomic regions in a different genetic background. To this end, backcrossing is a powerful resource in forward genetics. The evidence that experimental *L. major* backcrosses are possible, at least in vivo, suggests its use as a potential genetic tool for positional cloning and linkage analyses [12]. Thus far, the ability to produce backcross progeny is restricted to intraspecific crosses. Despite several attempts, backcrosses were not obtained when crossing the *L. major* × *L. infantum* F1 hybrids with the *L. major* parental line, leading to the conclusion that interspecies hybrids might be sterile [12]. It remains to be tested whether the same is true for other interspecies pairwise combinations in vivo or for irradiation-induced hybridization in vitro. Another challenge to be overcome in the in vitro hybrids in order to facilitate their use in future genetic studies is that all iHybrids described to date are polyploid, potentially restricting the number of genomic regions where parental genetic features can be isolated through recombination. Nonetheless, partial loss of heterozygosity was detected in F1 iHybrids in several parts of the genome leading to homozygous contribution of a parental genome [17]. It is hoped that as more effort is directed at manipulating the conditions of in vitro hybridization, the induction of a more complete meiotic program inclusive of the ploidy reduction steps and genome-wide recombinations between homologous chromosomes will be possible. 

Finally, the utility of experimental hybrids for QTL mapping requires that the readouts for phenotypes of interest are experimentally available. As the traits of greatest interest pertain to the diversity of clinical outcomes that different strains and species of *Leishmania* can produce, and which in certain cases can be modeled in a murine host, we have listed the crosses that we have carried out to date in sand flies and in vitro that are suitable for linkage analysis (Table 1).

## 6. Conclusions

Hybridization in *Leishmania*, although inferred by population genetic studies and formally demonstrated by the generation of hybrids in the laboratory, thus far lacks conclusive evidence that it is driven by a sexual process. Meiosis-like sexual recombination is supported by the presence of meiotic orthologs in the genome, and is the most likely system to account for the patterns of chromosome inheritance and recombinations revealed by bWGS of experimental hybrids generated in sand flies. Bi-parental inheritance of kDNA has been observed, although only the maxicircle kDNA from one parent appears to be retained during subsequent mitotic divisions. Progress in understanding the reproductive biology of *Leishmania* has been hampered by the need for both vector sand flies and genetically engineered parental strains to generate hybrids and the fact that *Leishmania* are facultatively sexual, yielding only a low frequency of hybrids in the crosses carried out to date. A major recent advance is the ability to generate stable hybrids entirely in vitro. By submitting culture promastigotes to DNA stress conditions, the frequency of hybridization and the ability to hybridize within and between different *Leishmania* species can be markedly enhanced. A transcriptionally unique population of hybridization-competent promastigotes expressing a number of meiotic gene orthologs has been identified following exposure to gamma radiation. Thus, with this expanded toolset any laboratory able to axenically grow *Leishmania* insect stages, i.e., virtually every *Leishmania* laboratory should now be able to generate recombinant parasites for genotype and phenotype analyses. Challenges remain in manipulating culture conditions to promote completion of their meiotic program, as the available in vitro hybrids are predominantly polyploid, in contrast to their diploid counterparts generated in sand flies. Further effort using in vitro protocols will hopefully lead to the identification and enrichment of the putative gametic cells and to direct observations of their meiotic cycle, including reductional division and the fusion of cells, nuclei, and kinetoplasts. Other outstanding questions related to genetic exchange could be addressed as well, including the inheritance of extrachromosmal circular DNA encoding drug resistance and the transmission or purging of intrinsic *Leishmania* RNA viruses. Perhaps most promising is the exploitation of in vitro hybridization for forward genetic analysis to map genes controlling the extraordinary diversity of the genus with respect to the range of infected hosts, tissue involvement, and pathologies associated with different strains and species of *Leishmania*. 

## Figures and Tables

**Figure 1 pathogens-11-00580-f001:**
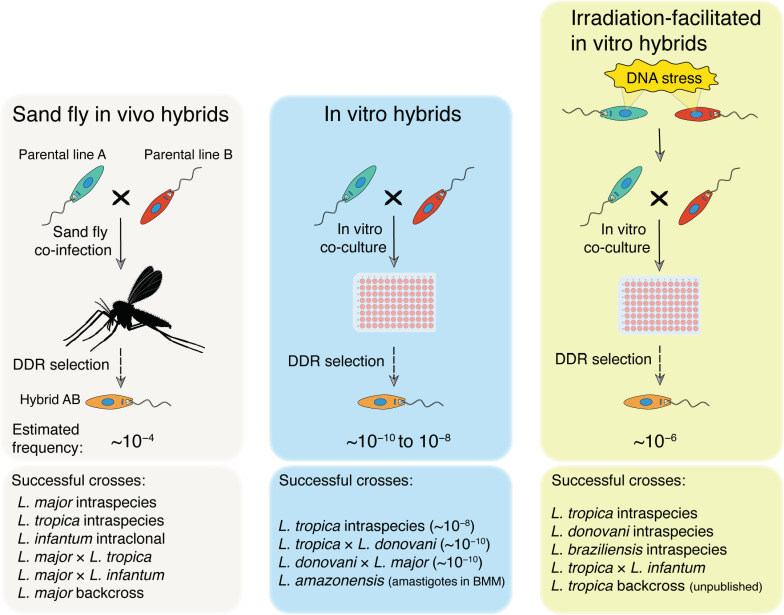
Experimental *Leishmania* hybridization. Three major methodologies for generation of *Leishmania* hybrids have been described using two drug-resistant parental promastigote cell lines: (1) sand fly co-infection [9,10,11,12,13] (**left** panel); (2) in vitro co-culture [15,17] (**middle** panel); and (3) in vitro co-culture of gamma-radiation treated promastigotes [16] (**right** panel). DNA stress may be induced by H_2_O_2_ or methyl methanosulfonate (MMS) treatment to increase mating competency. In all cases, hybrids were selected by double drug resistance (DDR) in culture. Estimated average minimum frequencies of mating-competent cells are shown for each method. Successful experimental crosses described using each method are listed in the bottom part of each panel. BMM: mouse bone marrow-derived macrophages.

**Table 1 pathogens-11-00580-t001:** *Leishmania* crosses and disease models in mice currently available for linkage studies.

*Leishmania* Cross	Parental Disease Phenotype in Humans	Parental Disease Phenotype in Mouse Models
*L. major* Fn × *L. major* Sd [11]	Healing cutaneous lesion Chronic cutaneous lesion [60]	C57Bl/6: healing ear dermal lesion [61] C57Bl/6: non-healing dermal lesion [61]
*L. major* Fn × *L. infantum* LLM-320 [13]	Healing cutaneous lesion Visceral leishmaniasis	C57Bl/6: healing ear dermal lesion; poor dissemination to or growth in the spleen [13] C57Bl/6: no dermal pathology; good dissemination to and growth in the spleen [13]
*L. major* Fn × *L. tropica* L747 (unpublished)	Healing cutaneous lesion Chronic cutaneous lesion	BALB/c: ear dermal lesion BALB/c: no dermal pathology (unpublished)
*L. donovani* SL2706 × *L. donovani* Mongi [16]	Cutaneous lesion [62] Visceral leishmaniasis	BALB/c: ear dermal lesion; no dissemination to or growth in the spleen [62] BALB/c: no dermal pathology; good dissemination to and growth in the spleen

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
