# Peer review of "Experimental Hybridization in Leishmania: Tools for the Study of Genetic Exchange"

_pathogens, 2022, doi:10.3390/pathogens11050580_

Round 1

Reviewer 1 Report

I want to congratulate the authors for the quality of the paper. The paper reviews studies about the hybridization of Leishmania and the possible implications in the parasite-host relationship.

Author Response

We appreciate the Reviewer's comment.   

Reviewer 2 Report

This review by Ferreira and Sacks deals with recent advances in our understanding of the phenomenon of genetic exchange in Leishmania. They also describe tools available to investigators to generate hybrids in in vitro culture settings.

This manuscript is clearly written and provides a very thorough overview of the litterature on the topic of genetic exchange and generation of hybrids. 

One aspect to clarify is the Leishmania strains amenable to genetic exchange studies. So far, all studies have been performed with lab-adapted strains. It would be of interest to discuss the potential of using clinical isolates with different phenotypes in the study of genetic exchange.  Are clinical isolates expected to generate hybrids as easily as lab-adapted strains? 

Another point that could be mentionned is the extrachromosomal circles amplified in drug-resistant Leishmania.  What is known about their fate during genetic exchange?

Author Response

One aspect to clarify is the Leishmania strains amenable to genetic exchange studies. So far, all studies have been performed with lab-adapted strains. It would be of interest to discuss the potential of using clinical isolates with different phenotypes in the study of genetic exchange.  Are clinical isolates expected to generate hybrids as easily as lab-adapted strains? 

This point is referred to on p. 3 of the text:   The relatively low frequency of hybrid recovery may be related to the use of culture adapted, recombinant parasites for hybrid selection, and it is unknown if fresh, clinical isolates might hybridize with greater efficiency in the vector.  

Another point that could be mentionned is the extrachromosomal circles amplified in drug-resistant Leishmania.  What is known about their fate during genetic exchange?

This point, along with the inheritance of Leishmania RNA viruses, is referred to in the concluding paragraph,  p. 12 of the text:   

Other outstanding questions related to genetic exchange could be addressed, including the inheritance of extrachromosmal circular DNA encoding drug resistance and the transmission or purging of intrinsic Leishmania RNA viruses.  

Reviewer 3 Report

This manuscript reviews state of the art in relation to experimental hybridization of Leishmania parasites. It is well written and clearly outlines and discusses current knowledge as regards the experiments done in sandflies, in vitro hybridization experiments and approaches into molecular mechanisms involved in meiosis. The review constitutes a valuable resource in the field.

It is however important to draw the attention of the authors about some facts to update or to discuss.

  • Kinetoplast DNA inheritance section should refer to more recent literature than ref 27. In Leishmania, each minicircle encodes only for 1 gRNA (and not 1-4). See for instance (Urrea et al., 2019; Simpson et al., 2015).
  • Mosaic aneuploidy in Leishmania is a constitutive genetic feature first described by Sterkers team (different papers including Lachaud et al., 2014), and is again brought to light by recent single cell sequence analyses (Negreira et al., 2022). How would this feature interfere with in vitro (or natural) hybrid formation? Would it explain some of your observations?

Line 205: please correct the typo in “putative”.

Author Response

Kinetoplast DNA inheritance section should refer to more recent literature than ref 27. In Leishmania, each minicircle encodes only for 1 gRNA (and not 1-4). See for instance (Urrea et al., 2019; Simpson et al., 2015).

These references have been added.  

Mosaic aneuploidy in Leishmania is a constitutive genetic feature first described by Sterkers team (different papers including Lachaud et al., 2014), and is again brought to light by recent single cell sequence analyses (Negreira et al., 2022). How would this feature interfere with in vitro (or natural) hybrid formation? Would it explain some of your observations?

Mosaic anuploidy is discussed on p. 5 of the text: 

The bWGS analysis of the parents and hybrid clones also revealed the copy number of the individual chromosomes and the patterns of somy inheritance.  While Leishmania are largely diploid, all Leishmania strains show varying degrees of aneuploidy, including mosaic aneuploidy, which refers to somy variations that are present within clonal populations23.  Thus, the analysis of somy at the population level, as was done in the published studies, computed average values.  In almost every case, chromosomes that were present in close to 3 copies in either parent were transmitted to the hybrid progeny in either 1 or 2 copies, and in frequencies expected under meiosis.  

Line 205: please correct the typo in “putative”.   Corrected